# BACH1 as a key driver in rheumatoid arthritis fibroblast-like synoviocytes identified through gene network analysis

Aurelien Pelissier[1,2,*], Teresina Laragione[3,*], Carolyn Harris[3], María Rodríguez Martínez[1,†] , Percio S Gulko[3,†]

RNA-sequencing and differential gene expression studies have significantly advanced our understanding of pathogenic pathways underlying rheumatoid arthritis (RA). Yet, little is known about cell-specific regulatory networks and their contributions to disease. In this study, we focused on fibroblast-like synoviocytes (FLS), a cell type central to disease pathogenesis and joint damage in RA. We used a strategy that computed sample-specific gene regulatory networks to compare network properties between RA and osteoarthritis FLS. We identified 28 transcription factors (TFs) as key regulators central to the signatures of RA FLS. Six of these TFs are new and have not been previously implicated in RA through ex vivo or in vivo studies, and included BACH1, HLX, and TGIF1. Several of these TFs were found to be co-regulated, and BACH1 emerged as the most significant TF and regulator. The main BACH1 targets included those implicated in fatty acid metabolism and ferroptosis. The discovery of BACH1 was validated in experiments with RA FLS. Knockdown of BACH1 in RA FLS significantly affected the gene expression signatures, reduced cell adhesion and mobility, interfered with the formation of thick actin fibers, and prevented the polarized formation of lamellipodia, all required for the RA destructive behavior of FLS. This study establishes BACH1 as a central regulator of RA FLS phenotypes and suggests its potential as a therapeutic target to selectively modulate RA FLS.

## Introduction

Rheumatoid arthritis (RA) is a common systemic autoimmune and inflammatory disorder characterized by synovial inflammation and hyperplasia that may lead to joint destruction (1, 2, 3). New biologic disease-modifying anti-rheumatic drugs (bDMARDs) and JAK inhibitors that target various inflammatory pathways have significantly improved disease control and outcomes (4, 5). Yet, a considerable number of RA patients still have only partial or no response to therapy, and sustained remission remains uncommon (6). The development and progression of RA involves dynamic interactions between multiple genetic and environmental factors, and therefore, understanding the heterogeneous pathophysiological processes in RA patients remains a major challenge (7). Among the cell types found in synovial tissues, fibroblast-like synoviocytes (FLS) (8, 9) are centrally relevant in the RA pathogenesis (8, 10, 11). In RA, FLS become activated producing increased levels of inflammatory mediators that contribute to driving the local inflammatory response and joint damage (12). FLS also have increased invasive properties that together with their increased production of proteases contribute to cartilage and bone damage (8). Although past studies have linked the invasiveness of FLS with the severity of joint damage, both in RA patients (13) and in rodent models (14), the underlying mechanisms of those processes remain incompletely understood.

Genome-wide association studies and differential gene expression (DEG) studies have significantly improved our understanding of the disease's genetic underpinnings (15, 16). Yet, those studies did not differentiate the contribution of individual cell types to disease. In this context, mapping the identified transcriptional and immune signatures specific to FLS could substantially expand our understanding of RA disease processes (17). Identifying transcription factors (TFs) and their associated regulatory signatures is of particular interest, as TFs play a pivotal role in regulating gene expression (18). Moreover, with an estimated count of 1,000 TFs in humans (19), identifying those that govern the phenotypic traits of FLS in RA may open new possibilities for novel therapeutic target discovery. Recently, single-cell RNA-sequencing studies are offering valuable insights into understanding diseases

[1]IBM Research Europe, Eschlikon, Switzerland   [2]Department of Biosystems Science and Engineering, ETH Zurich, Basel, Switzerland   [3]Division of Rheumatology, Icahn School of Medicine at Mount Sinai, New York, NY, USA

Correspondence: maria.rodriguezmartinez@yale.edu; percio.gulko@mssm.edu
Aurelien Pelissier's present address is Institute of Computational Life Sciences, ZHAW, Winterthur, Switzerland
María Rodríguez Martínez's present address is Yale School of Medicine, New Haven, CT, USA
*Teresina Laragione and Aurelien Pelissier contributed equally to this work
†María Rodríguez Martínez and Percio S Gulko contributed equally to this work

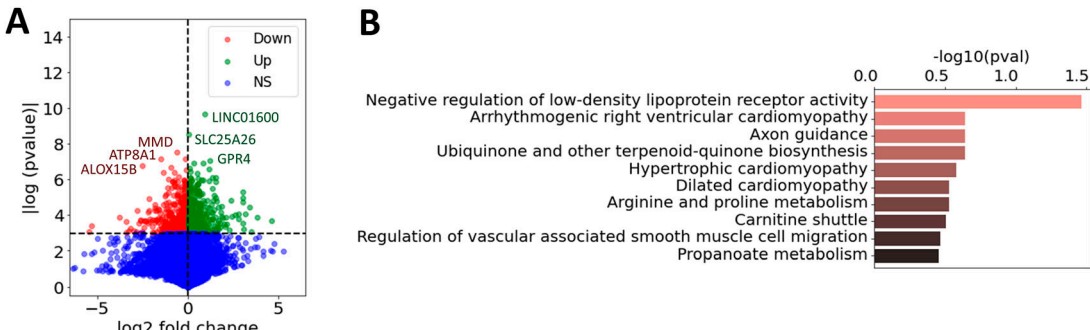

**Figure 1. Differentially expressed genes in RA versus OA FLS.**
**(A)** Volcano plot of the computed $\log_2$ fold change and *P*-values for all the expressed genes in RA versus OA fibroblast-like synoviocytes (*t* test), where the six genes with the highest *P*-values were marked. Genes were labeled as down-regulated in RA, up-regulated in RA, or non-significant (NS). **(B)** Most enriched pathways of the differentially expressed genes in RA versus OA fibroblast-like synoviocytes (1,093 genes with $t_{\text{diff-expr}} > 2$), combined from GO (28), KEGG (29), and Reactome (30), ranked by adjusted *P*-values.

at a cell-specific level (20, 21). Still, using these datasets presents several challenges because of limited patient numbers, batch effects, and sparse data (22). Consequently, inferring networks based on these datasets has proven to be particularly difficult (23, 24, 25).

In this study, we provide a comprehensive analysis of gene regulation in RA FLS. We used a cell type–specific bulk RNA-seq dataset of FLS from patients with RA (n = 18) and OA controls (n = 13) (20) to construct sample-specific gene regulatory networks (GRNs) for both RA and control samples, generating 18 RA and 12 OA FLS networks. Employing sample-specific GRNs is essential for capturing the heterogeneity of RA, as it tailors the networks to each individual sample. This contrasts with previous studies, which focused on cohort-specific GRNs (7, 26, 27), allowing us to preserve intersample variability. Through differential analysis of the network edges, we identified key transcription factors driving FLS-specific expression differences, offering new insights into RA. Among the most prominent and specific regulators in FLS were BTB and CNC Homology 1 (BACH1) and H2.0-like homeobox (HLX). To validate our discovery strategy, we conducted experiments silencing BACH1 in FLS cell lines from RA patients, demonstrating its influence on FLS migration, adhesion, and lamellipodium formation. Our study is an important step toward the development of novel RA treatments with FLS-specific pathway targeting.

## Results

### Differentially expressed genes and pathways in RA versus OA

We used cell type–specific RNA-seq data from FLS (20) to obtain gene expression profiles for 18 RA and 12 OA biopsies. Like in most RA FLS studies, OA samples were used as controls given the lack of available normal biopsies. Next, we computed the differential gene expression in each dataset (*t* test) between RA and OA samples. This means that for each gene, we obtain a differential expression score (denoted $t_{\text{diff-expr}}$) from which we can extract a set of DEGs (Table S1). The genes with the most significant difference included the long intergenic non-protein coding RNA 1600 (LINC01600), the solute carrier family 25 member 26 (SLC25A26), GPR4, and monocyte-to-

macrophage differentiation–associated (MMD; Fig 1A). Pathway analyses were done including 1,093 genes with a $t_{\text{diff-expr}} > 2$, and revealed an overrepresentation of "negative regulation of low-density lipoprotein receptor activity," followed by others such as axon guidance and carnitine shuttle (Fig 1B, Table S2).

### Key TF regulators and pathways driving regulatory differences between RA and OA FLS

Gene expression phenotypes are more effectively interpreted through the identification of the TFs that regulate them. Therefore, we next identified the TFs regulating RA phenotypes by inferring sample-specific GRNs and identifying differential regulatory patterns between RA and OA. We aimed to get insight into the regulatory relationships between DEGs, and whether they involved TF-mediated inhibition or activation, and co-regulations. To explore this question, we constructed 18 RA and 12 OA FLS-specific GRNs by integrating RA gene expression datasets (20) with prior knowledge about TF binding motifs (from the Catalog of Inferred Sequence Binding Preferences, CIS-BP (31)) and about protein–protein interactions (from StringDB (32)). Briefly, we leveraged LIONESS (33) to compute individual GRNs for each biopsy sample (see the GRNs in FLS section, Fig 2A). These networks incorporate edge weights to quantify the likelihood of regulatory interactions between TFs and their target genes (TGs). We then compared these edge weights between RA and OA samples using a *t* test and obtained a score $t_{\text{diff-edge}}$ for each edge TF-TG. From these edge-specific differential analyses, we generated a novel network, the *differential GRN* (dGRN), which illustrates the differential regulation between RA and OA (Fig 2B). To quantify TFs' regulatory function, we calculated a TF regulatory score ($t_{\text{reg}}$). This score is calculated based on the average absolute differential weight of the regulatory edges between each TF and its associated TGs, that is, the average of $|t_{\text{diff-edge}}|$ between a TF and its targets (see the Analysis of TF RA regulatory activity in GRNs section).

We identified 185 TF signatures in RA FLS, and show the top 20 ranked by their regulatory scores (Table 1, with Z-statistics of each TF in parentheses). The Z-statistic quantifies the number of standard deviations by which the score of a given TF deviates from the mean score across all TFs. For additional insight into the important pathways

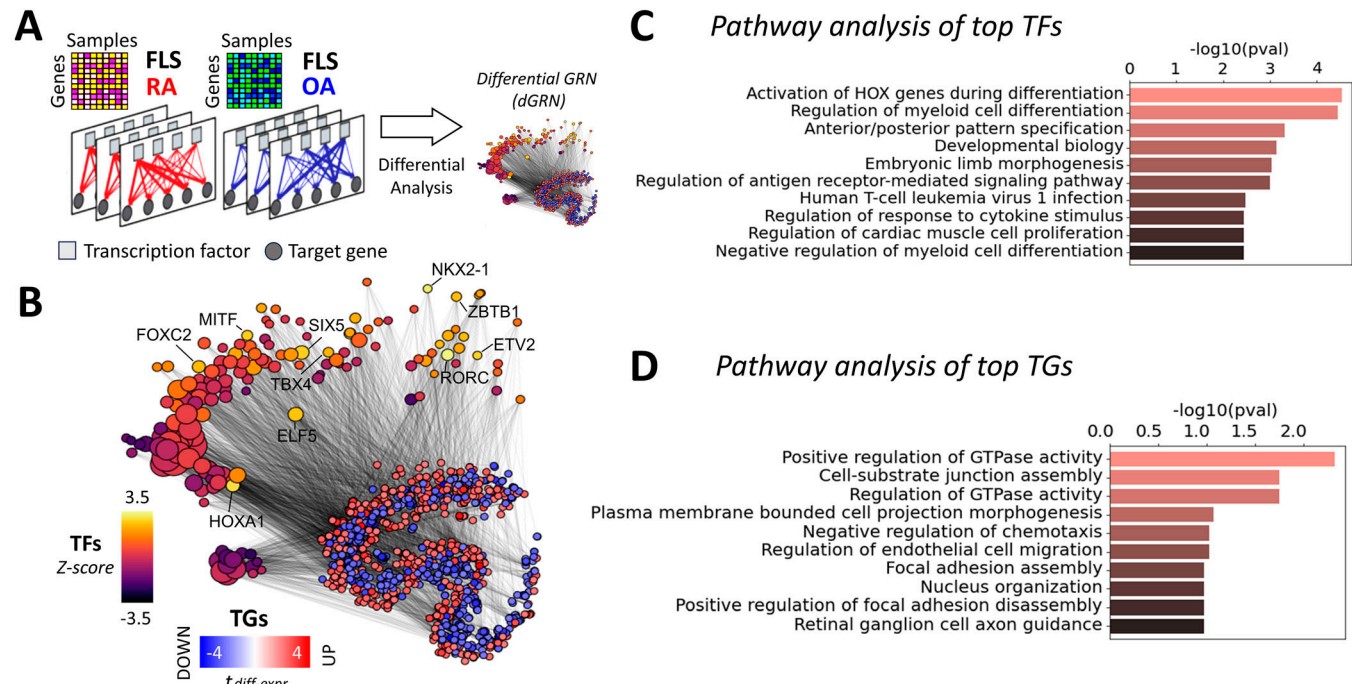

**Figure 2. Differential gene regulatory network of RA FLS.**
**(A)** 18 RA and 12 OA fibroblast-like synoviocyte (FLS) networks were inferred from FLS gene expression profiles (20). Networks incorporate edge weights representing the probability of regulatory interactions between transcription factors (TFs) and their target genes (TGs). The differential analysis of the edge weights between RA and OA resulted in the assembly of a *differential GRN* (dGRN) specific to RA FLS. **(B)** t-SNE representation of the dGRN of RA FLS. For visual clarity, only TGs with a differential expression score in FLS above 2 ($t_{diff-expr} > 2$), and only edges with a differential edge weight score above 2 ($t_{diff-edge} > 2$), are shown. TF node sizes are proportional to the node degree and colored according to their regulatory score in the network. Gene nodes are colored according to their differential expression score in FLS. The top 10 TFs ranked by their regulatory scores are labeled. **(C)** Top 10 enriched pathways of the major TFs involved in RA FLS, compiled from GO (28), KEGG (29), and Reactome (30), and ranked by their adjusted *P*-values. **(D)** Top 10 enriched pathways of the major TGs involved in RA FLS, ranked by adjusted *P*-values.

**Table 1. Top 20 ranked TF regulators in fibroblast-like synoviocytes with their Z-statistics provided in parentheses.**

| Ranks 1–5 | Ranks 6–10 | Ranks 11–15 | Ranks 15–20 |
|---|---|---|---|
| RORC (3.39) | SIX5 (2.61) | CBFB (2.16) | ETV3 (2.04) |
| NKX2-1 (3.37) | ELF5 (2.46) | BBX (2.16) | TLX3 (1.97) |
| HOXA1 (3.20) | ZBTB1 (2.42) | TAL1 (2.13) | HOXB3 (1.95) |
| ETV2 (2.67) | FOXC2 (2.27) | ZBTB3 (2.11) | PKNOX1 (1.95) |
| MITF (2.64) | TBX4 (2.23) | RARG (2.09) | MECOM (1.94) |

involved in RA FLS, we selected the 185 TFs with a Z-statistic above 0.5 (Table S3). This collection constitutes our set of key *TF drivers* in RA FLS. Using this set, we performed a pathway enrichment analysis using the Gene Ontology (GO) (28), KEGG (29), and Reactome (30) databases (Fig S1). Given that all TFs are DNA-binding proteins, we removed any terms associated with RNA and DNA transcription, as these processes are ubiquitous and therefore not likely to be specific to RA. After this filtering, 77 significantly enriched (adjusted $P < 0.05$) pathways were identified. The 20 most significant pathways included HOX gene activation, regulation of myeloid cell differentiation, and TNF signaling, among others (Fig 2C, Table S4).

In addition to evaluating the TF regulation of RA, we also investigated TGs that were differentially targeted by more than one of the key TF drivers we identified (185 TFs with Z-statistics above 0.5). Interestingly, we found that some TGs were differentially targeted ($t_{diff-edge} > 1$) by more than 30% of these TFs (the top targeted genes were ALDH1A2, SYNE2, and NF1). We provide the top 100 targeted genes in Table S5. A pathway enrichment analysis of the 211 TGs targeted by more than 20% of the key TF drivers in RA highlighted the pathways "positive regulation of GTPase activity," "Cell–substrate junction assembly," and "negative regulation of chemotaxis," among others (Fig 2D, Table S6).

## An independent approach increases the confidence in the identified regulators

Relying on the predictions of a single computational method might lack the robustness required to identify promising therapeutic targets. To increase our confidence in the identified RA regulators, we augmented our study by incorporating a selection of preexisting literature-derived networks, which also included edge weights as a metric for assessing the confidence of the interactions between nodes. These include RIMBANET (34), StringDB (32), and GIANT (35), a collection of networks that accurately capture tissue-specific and cell type–specific functional interactions. As RA is an autoimmune disease, we selected networks computed from immune-related tissues and cell types (including lymph nodes,

spleen, tonsils, and blood). In summary, 14 additional networks were collected for our analysis, as detailed in the Table S7.

Although these networks recapitulate general immune knowledge derived from various data types, they are not specific to synovial tissues. Therefore, they are unable to discern RA-specific relationships between TFs and TGs as effectively as the PANDA framework does. We hence designed a different approach based on the *key driver analysis* (KDA) (36), a computational pipeline to uncover major disease-associated regulators or causative hubs in a biological network (see the KDA section). Briefly, genes exhibiting more connections to RA-associated genes than expected by random chance were considered potential drivers. KDA analysis requires the definition of RA-associated signatures, that is, lists of genes associated with the disease. Here, we compiled known RA-

associated genes from the literature, including genome-wide association studies (16, 37, 38), knowledge-based datasets (39, 40, 41), and known drug targets (42, 43), as well as prior meta-studies and datasets (44, 45, 46, 47, 48) (see the KDA section). Using this list of RA-associated signatures as a starting point, we obtained a list of 174 key TF drivers that were identified as a *key driver gene* in at least one of the 14 networks (Table S8).

A final list of potential key TF regulators in FLS was generated by combining our dGRN with the KDA approach (Table 2); namely, we retained TFs (i) whose regulatory scores ($t_{reg}$) computed with the dGRN network exceeded half an SD above the mean score of all TFs (i.e., Z-statistics > 0.5), and (ii) that were identified as key driver genes by the KDA approach described above. We identified 28 TFs that met these two conditions. Among them, 7 were new TF

**Table 2. 28 key TFs implicated in the regulation of fibroblast-like synoviocytes (FLS) in RA, as identified in our analyses, are ranked by Z-statistics.**

| TF | Synovial cell types[a] | FLS score (dGRN Z-statistics) | KDA[b] (Lit-DEG) | References (in vivo/ex vivo studies) |
|---|---|---|---|---|
| MITF | FLS, T cells | 2.64 | 6–4 | (50) |
| CBFB | FLS | 2.16 | 2–1 | Asso. with RUNX1 (51) |
| IKZF1 | FLS | 1.93 | 6–6 | (52 Preprint) |
| HLX | FLS | 1.90 | 5–5 | None |
| BACH1 | FLS | 1.83 | 6–1 | None |
| FOS | FLS | 1.72 | 8–3 | (53) |
| ETV7 | FLS, T cells | 1.57 | 2–2 | None |
| RFX5 | FLS, monocytes, T cells | 1.54 | 4–3 | (54) |
| HIF1A | FLS, monocytes | 1.52 | 9–4 | (55) |
| TGIF1 | FLS, T cells | 1.39 | 8–4 | None |
| ELF4 | FLS, T cells | 1.34 | 9–7 | (56) |
| RUNX1 | FLS | 1.27 | 8–4 | (57, 58) |
| NFATC1 | FLS | 1.23 | 3–3 | (59, 60) |
| IRF7 | FLS, monocytes | 1.18 | 7–4 | (61) |
| CREM | FLS, B cells, T cells | 1.16 | 6–2 | (62) |
| FOSL1 | FLS, B cells, T cells | 1.12 | 1–1 | (63) |
| FLI1 | FLS | 1.09 | 9–8 | (63) |
| ELF1 | FLS, T cells | 1.03 | 2–2 | None |
| BHLHE40 | FLS, B cells | 0.98 | 8–1 | (64) |
| STAT1 | FLS | 0.87 | 12–10 | (65) |
| EGR2 | FLS, B cells | 0.83 | 6–5 | (66) |
| NR4A1 | FLS, T cells | 0.75 | 3–1 | (67) |
| FOSL2 | FLS | 0.71 | 9–3 | (68) |
| JUNB | FLS | 0.68 | 9–8 | (69) |
| HIVEP1 | FLS, T cells | 0.65 | 2–1 | None |
| PLAGL1 | FLS, B cells, T cells | 0.62 | 1–1 | None |
| ENO1 | FLS | 0.59 | 3–1 | (70) |
| HOXB2 | FLS | 0.54 | 1–2 | (71) |

A higher Z-statistic indicates a greater regulatory implication in FLS signatures.
[a]Some of the key TFs in FLS were also found as key regulators in other synovial cell types (49).
[b]The left and right numbers correspond to the number of networks, among 14, in which the TF was identified as a KDG using, respectively, the literature and DEG list (49).

regulators (HLX, BACH1, ETV7, TGIF1, ELF1, HIVEP1, and PLAGL1) not previously implicated in RA. The remaining TFs had been previously reported in RA studies, further validating our discovery strategy (Table 2). By integrating our results with those from a parallel study on other cell types (49), which employed a similar methodology, we found that only BACH1 and HLX were specific to FLS, whereas the other factors were also associated with T- and/or B-cell signatures. Importantly, the TFs shown in Table 1 are quite different from those in Table 2. Incorporating condition (ii) reduced our previous list from 185 TFs to 28 TFs. With this strategy, we increased the confidence in our results, as the chances of selecting false positives were reduced by combining the two criteria.

## TF-TF co-regulation network in FLS

Having identified 28 significant TFs associated with RA in FLS, we next evaluated their co-regulatory activity, potentially revealing distinct clusters of co-regulation. TF-TF co-regulation was quantified using the Pearson correlation between the regulatory differential activities of the TFs and their TGs, that is, $corr(t_{\text{diff-edge}}(TF_i), t_{\text{diff-edge}}(TF_j))$, focusing only on the TGs that are common to both TFs (see the TF-TF co-regulation network section). We used a hierarchical clustering algorithm to identify groups of co-regulatory TFs using a correlation threshold of 0.5. This analysis identified 5 TF clusters, with four small and one major group containing 17 TFs (Fig 3A). This suggests a major co-regulated hub in RA FLS, with BACH1, HIF1A, TGIF1, and FOSL1 having a central key driving role (highest Z-statistics among TFs in the largest cluster). Among these, BACH1 has the highest Z-statistics (Table S9, Fig 3A). These clusters were not completely independent, as there was also co-regulatory activity across TF belonging to different clusters (Fig 3B).

## BACH1 regulatory network in RA FLS

BACH1 was among the strongest TF regulators identified in RA FLS. To examine the regulatory effect of BACH1 on FLS, we constructed the network of the BACH1 target genes that were also DEGs between RA and OA (Fig 4A). This network included other significant TFs, such as PAX8, CBFB, NFE2L2, ETS1, RUNX1, and SMAD4, which also contributed to the co-regulation of major BACH1 targets. Interestingly, RUNX1 and CBFB were also identified as significant TF drivers in RA FLS, but in a distinct co-regulatory cluster, with only moderate correlations to BACH1 (0.31 and 0.16, respectively). This suggests that although these two TFs are targets of BACH1, they exert control over their respective targets independently of BACH1 regulation.

Besides these TFs, this network also included 131 genes that were differentially targeted by BACH1 (Table S10). A pathway enrichment analysis of these genes showed an overrepresentation of the "fatty acid degradation" and "ferroptosis" pathways, driven by genes HADHB, ACSL6, and CPT1C, among others (Fig 4B and Table S11). These findings are in line with the literature, where it has been reported that BACH1 promotes ferroptosis (72) and suppresses fatty acid biosynthesis (73). These observations implicate for the first time BACH1 in the regulation of RA FLS metabolism and ferroptosis.

## Effect of BACH1 knockdown on gene expression

To gain more direct insight into the role of BACH1 in regulating RA FLS, we ranked the BACH1 regulatory weights (average likelihood of TF-TG interaction across all constructed FLS networks in OA and RA) to its TGs into four separate quartiles (Q1–Q4, Fig S2). We anticipated that the expression of TGs with a high regulatory weight (Q4) would be more significantly affected by a BACH1 knockdown than those with a low regulatory weight (Q1). To investigate this hypothesis, we used siRNA to knock down BACH1 in RA FLS cell lines developed from RA patients and performed RNA sequencing to assess the resulting changes in gene expressions (see the siRNA knockdown section). We identified 24 up-regulated and 14 down-regulated genes between the two groups (t test, $P < 0.05$ after the Benjamini–Hochberg correction (74), Fig 5A, Table S12). Among these, BACH1 exhibited significant down-regulation with a $\log_2$ fold change of –2.5 (adjusted $P$-value of $5.12 \times 10^{-6}$), confirming the effectiveness of our knockdown. Interestingly, BACH1's target genes with the lowest edge weight (Q1) had a significantly lower measured fold change compared to those with the highest weights (Q4) (0.3 versus 0.5, $P = 3.46 \times 10^{-7}$, t test), indicating a good agreement between the measured gene expression fold changes with the regulatory edge weights of the BACH1 FLS networks (Fig 5B), further validating the RA FLS network we constructed.

Among the genes with the most significant increase in mRNA levels after siRNA BACH1 knockdown were HMOX1, RP11-863P13.3 and RP11-863P13.4 (two lincRNAs), ZNF469, and TRIB3, and among those with reduced expression were SLC35A5 (a nucleoside-sugar transporter), NUDT15, CBFB (a TF), and STRADB (involved in cell polarity and energy-generating metabolism). Pathway analyses of the identified DEGs increased the representation of genes implicated in "intrinsic apoptotic signaling pathways in response to endoplasmic reticulum (ER) stress," "response to ER stress," and "response to EIF2AK1" (Fig 5C and Table S13). BACH1 is implicated in oxidative stress (75), and oxidative stress may have a role in ER stress.

## Effect of BACH1 knockdown on FLS migration, adhesion, lamellipodium formation, and cell morphology

We next examined the role of BACH1 in RA FLS behaviors relevant to disease and joint damage (see the FLS assays section). siRNA knockdown of BACH1 significantly reduced RA FLS adhesion by an average of 50% ($P = 0.021$), and reduced FLS migration in the wound healing (scratch) assay by ~40% ($P = 0.0067$, paired t test, Fig 6A). FLS cell morphology was analyzed under immunofluorescence microscopy, showcasing actin fibers and lamellipodia, marked by phalloidin and pFAK staining, respectively (see the Immunofluorescence microscopy section, Fig 6B and C). Images were obtained from 20 cells per FLS cell line (a total of four different cell lines, each from a different RA patient). Knockdown of BACH1 significantly affected FLS morphology, reducing the number of cells with thick and polarized actin fibers, as well as reducing the number of cells with an elongated shape (Fig 6A and B). Knockdown of BACH1 also significantly reduced the numbers of pFAK-positive lamellipodia, which has a critical role in

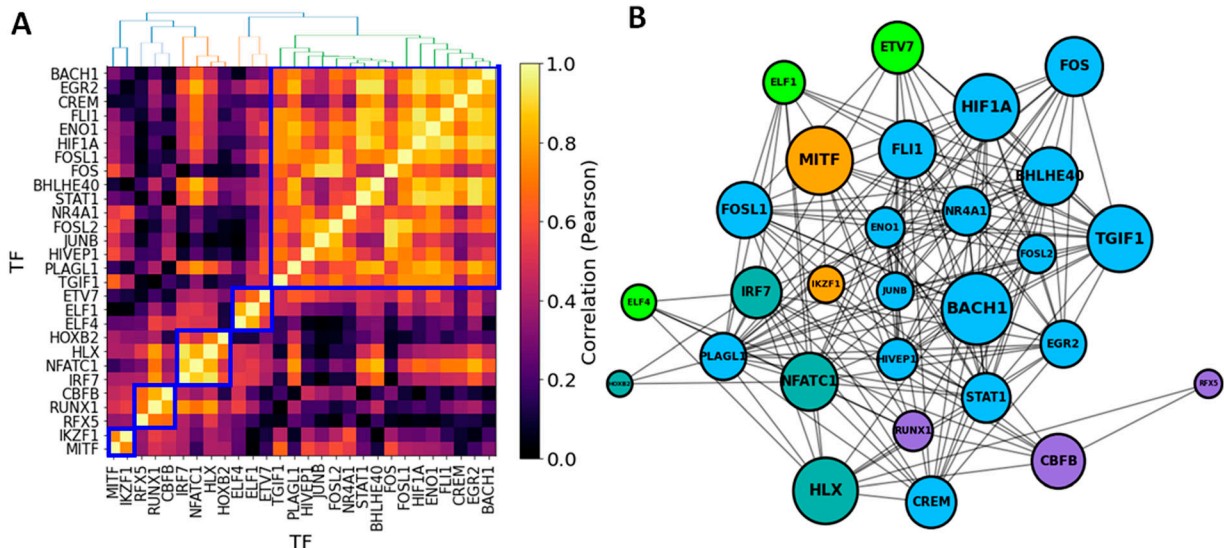

**Figure 3. Co-regulatory activity of significant TFs in RA fibroblast-like synoviocytes.**
**(A)** Pairwise TF-TF co-regulation heatmap, quantified in terms of the Pearson correlation between the differential edge weight $t_{\text{diff-edge}}$ and their common TGs (see the TF-TF co-regulation network section). A hierarchical clustering approach was used to group the TFs into clusters (depicted in blue square). **(B)** Network visualization of the major TFs involved in fibroblast-like synoviocyte regulation in RA. Edges represent a correlation > 0.5, and node sizes are proportional to both the degree and the TF regulatory scores ($t_{\text{reg}}$). Nodes are colored according to their cluster assignment.

**Figure 4. BACH1 regulatory network in RA FLS.**
**(A)** Regulon of BACH1 includes many genes that are differentially expressed in fibroblast-like synoviocytes. BACH1 also interacts with 6 TFs (green squares), and in a tight cluster that contributes to the co-regulation of BACH1 target genes. Interactions between TFs are shown with a red edge, whereas TF-TG interactions are shown with a gray edge. Edge thickness is proportional to edge weights, node sizes are proportional to RA differential t-scores ($t_{\text{diff-edge}}$), and target nodes are colored according to their RA differential expression in fibroblast-like synoviocytes ($t_{\text{diff-expr}}$). **(B)** Top 5 overrepresented pathways identified among the BACH1 differentially expressed target genes.

movement and adhesion (76). The FLS production of total or activated matrix metalloproteinases (MMP2 and MMP3) and cell invasion through Matrigel were not significantly affected by BACH1 knockdown (see the Matrix metalloproteinase [MMP] quantification in zymograms section, Figs 6A and S3).

## Discussion

Sustained disease remission is still rarely achieved in RA (78). Available treatments, including biologics and JAK inhibitors targeting different aspects of the immune response, achieve similar rates of response (4, 79). FLS have a central role in RA pathogenesis (8), driving leukocyte chemotaxis into the synovial tissues, and mediating bone and cartilage damage (8). The RA FLS have a highly invasive and destructive behavior that correlates with joint damage (80). However, the FLS subtypes, states of activation, and their characteristics are only now beginning to be comprehended (11, 20, 21, 81), potentially paving the way for the identification of novel targets for the development of new treatments aiming to achieve sustained remission, while minimizing the risk of immunosuppression in RA patients. The recent availability of large transcriptomic studies and datasets has also expanded our understanding of RA pathogenic processes. Yet, the intricate molecular and cellular pathways, along with their regulation and interactions, remain largely elusive. Gene regulatory processes vary across cell types, and only a limited number of studies have characterized RA drivers in cell types relevant to RA pathogenesis, such as FLS (11, 18, 82, 83).

In this study, we leveraged FLS RNA gene expression data (20) to infer FLS-specific GRNs, and created one for each synovial biopsy in the datasets. Then, a differential analysis of these networks enabled a comparison between RA versus OA and revealed key

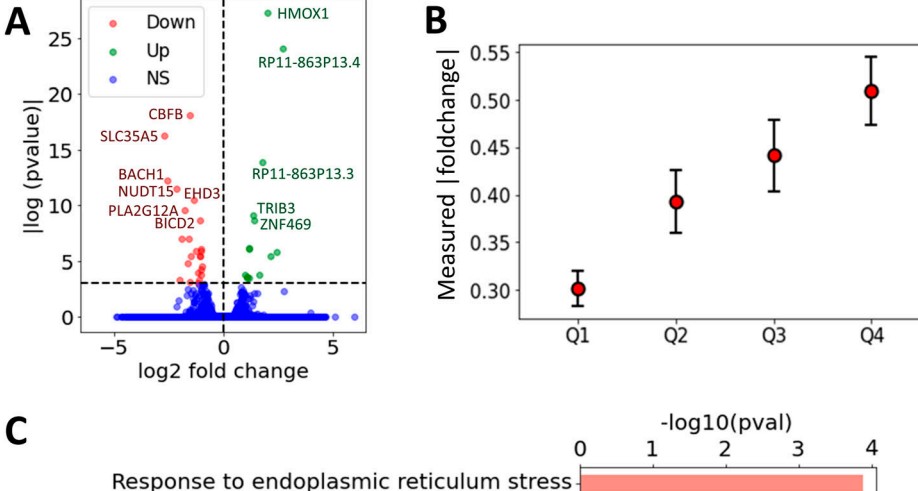

**Figure 5. Effect of BACH1 knockdown on gene expression in RA FLS.**
**(A)** Volcano plot of the measured log$_2$ fold change and *P*-values for all the expressed genes without and with BACH1 knockdown (paired *t* test), where the genes with a *P*-value below 0.001 were annotated. Genes were labeled as down-regulated, up-regulated, or non-significant (NS). **(B)** Measured log$_2$ fold change of gene expression between the siBACH1 and siCTL group, averaged over BACH1 target genes grouped by the four quartiles Q1–Q4. Errors bars represent the 95% confidence interval, defined as $std/Q_n$. **(C)** Top 10 overrepresented pathways identified among the differentially expressed genes with and without BACH1 knockdown.

pathways and gene interactions associated with each condition. Using sample-specific GRNs was crucial for capturing the heterogeneity of RA, as it allowed the networks to be tailored to each individual sample, thus contrasting to previous studies on FLS, that predominantly focused on cohort-specific GRNs (11, 18, 82, 83). In addition, we directly analyzed FLS from human synovial tissue biopsies, whereas these prior studies did not, potentially offering new insights into RA mechanisms. Here, we highlighted potential therapeutic targets and molecular markers that differentiate RA from OA. For example, we identified an overrepresentation of GO pathways involved in the regulation of GTPase activity in RA. Interestingly, GTPases such as Rac1 and RhoA have been implicated in the regulation of RA FLS behaviors including adhesion, migration, and invasion (84, 85, 86). As these networks contained regulatory information between TFs and their TGs, our analysis allowed us to rank the TFs according to their contribution to the observed differential gene expression between RA and OA (Table 1).

We further refined our findings using an alternative computational methodology (49) to enhance robustness and reduce the number of false positives. With this approach, we identified 28 candidate key driver TFs in FLS (Table 2), including seven not previously implicated in RA (BACH1, HLX, ETV7, TGIF1, ELF1, HIVEP1, and PLAGL1). Of those seven, only BACH1 and HLX were specific to FLS, whereas the others were also associated with T- and/or B-cell signatures (Table 2). Interestingly, we identified significant TF-TF co-regulated networks where BACH1 was the most significant driver. Among the genes differentially targeted by BACH1 were genes involved in fatty acid degradation and ferroptosis, an iron-dependent programmed cell death. BACH1 promotes ferroptosis, in part by repressing genes that interfere with iron-induced oxidative stress (72, 87). Ferroptosis has been implicated in cancer cell behavior and metastasis, but its

potential role in RA remains uncharacterized. BACH1-induced ferroptosis has been recently demonstrated to inhibit fatty acid biosynthesis (72, 73), and fatty acids, essential for normal FLS functioning, are not as crucial for RA FLS, which have reduced usage and reduced fatty acid beta-oxidation (88). BACH1 also regulates cancer cell metabolism beyond fatty acids, including the production of lactate via hexokinase 2 (HK2) (89). HK2 is central to RA FLS behaviors and rodent arthritis (90). Thus, BACH1 emerges as a novel regulator of both ferroptosis and metabolism in RA FLS.

Knockdown of BACH1 in RA FLS significantly increased mRNA levels of HMOX1, a gene transcriptionally suppressed by BACH1. Other genes expressed in increased levels in BACH1 knockdown FLS included lincRNAs, the CBFB, and STRADB (involved in cell polarity and energy-generating metabolism). Knockdown of BACH1 increased the expression of "intrinsic apoptotic signaling pathways in response to endoplasmic reticulum (ER) stress" and "response to ER stress" pathways. Moreover, cell ER stress has been linked to driving synovial inflammation (91). We examined the effect of siRNA knockdown on RA FLS phenotypes and observed that in the absence of BACH1, RA FLS were not able to take an elongated shape, and did not form thick actin fibers or lamellipodia. Without these morphological changes, FLS are unable to move, which is crucial for invasion. Indeed, BACH1 knockdown cells had a reduced ability to adhere and reduced mobility. Furthermore, BACH1's role in osteoclastogenesis (92), combined with the critical function of osteoclasts in RA-induced bone damage, underscores the potential advantage of BACH1 inhibition in RA therapy.

In addition to the BACH1 discovery, the highest ranked TFs in RA FLS included two homeobox genes (NKX2-1 and HOXA1, Table 1) not previously implicated in arthritis. Our integrated strategy identified additional key TFs driving the gene expression signatures in FLS,

**Figure 6. BACH1 silencing and changes in RA fibroblast-like synoviocytes (FLS).**
**(A)** BACH1 silencing reduced RA FLS adhesion and migration (wound healing assay), and affected cell morphology, including key characteristics required for movement and invasion such as thick and linear visible actin filaments (77), elongated shape, and the unidirectional formation of lamellipodia (bars represent the mean of each phenotype in each group; *P < 0.05, paired *t* test). Phenotype scores on the *y*-axis were quantified either by the % of cells with the described attribute, or by dividing the score by the highest measured value and multiplying by 100 (indicated as normalized). **(B, C)** Representative immunofluorescence microscopy images of RA FLS (magnification 500x), showing actin fibers and lamellipodia, marked by phalloidin and pFAK, respectively. **(B, C)** FLS were treated with (B) siRNA BACH1, showing a stellate morphology, with disorganized actin fibers and no lamellipodia, whereas (C) cells treated with siRNA control (CTL) had the typical RA FLS elongated morphology with thick and organized actin fibers, as well as polarized formation of lamellipodia.

and potentially new targets for functional studies including HLX, MITF, FOSL1, and ETV7, among others. Importantly, although our methodology differs significantly from previous studies on RA FLS (11, 18, 82, 83), in terms of both methods and datasets, many of the key drivers we identified were also highlighted in those works. Reference 11, which used epigenetic data and generated their own FLS cell line datasets, identified BACH1, FOSL1, and ETV7, consistent with our results. Reference 83 derived their network from scRNA-seq data on mice and mapped it to human data, with RUNX1 and BACH1 emerging as shared drivers. Reference 18 focused on DEGs from RA versus OA synovial tissues, though this approach is limited by the known cellular composition differences between these conditions (49). Nevertheless, they identified STAT1 and IRF4, two prominent RA transcription factors, which we also found. Lastly, reference 82 constructed a qualitative Boolean network based on prior RA FLS knowledge and, like us, identified HIF1. By integrating our findings with these studies, we can confidently assert that transcription factors such as BACH1, FOSL1, ETV7, RUNX1, STAT1, IRF4, and HIF1 represent therapeutic targets. Notably, our work provides a more in-depth analysis of BACH1, whereas the other studies only briefly mention it as a potential driver.

Finally, an intriguing aspect that we did not address in this article is the heterogeneous pathophysiological background of RA (7, 11).

There is considerable evidence of synovial tissue heterogeneity in RA, potentially contributing to the difference in therapeutic response to bDMARD therapy across patients (93). We could leverage our method to infer sample-specific GRNs to distinguish between the GRNs of different FLS subgroups. As FLS have emerged as a leading contender for the source of this RA heterogeneity (10, 94), such an approach might provide crucial insights into the regulation behind the observed heterogeneity of therapeutic responses in RA patients.

In conclusion, this work presents a novel and integrated methodology for analyzing RA FLS transcriptomic data. Our methodology led to the identification of new genes and pathways for further investigation and potential therapeutic targeting. We identified BACH1 as a new key TF driving the gene expression and phenotypic characteristics of RA FLS. Although BACH1 emerged as the primary driver in our analyses, our findings pave the way for additional investigations into the roles of various TFs in RA FLS and their co-regulation, and whether targeting one of them is enough to affect disease as BACH1 analyses suggest. In a broader context, the computational approach described in this article, with the use of statistical techniques to compare network properties across samples and phenotypic groups, has the potential to also effectively be used to analyze other synovial cell types, such as T cells, B cells, and monocytes (49), or be adapted to the study of other diseases.

# Materials and Methods

## Gene expression data and normalization

Our analysis leverages the cell type–specific bulk RNA-seq study of FLS from patients with RA (n = 18) and OA controls (n = 13) from the Accelerating Medicines Partnership (AMP) Phase I [20]. We did not use the scRNA-seq data from this study as they only provided FLS scRNA-seq expression from three patients. All data underwent scaling normalization [95] to remove potential biases of other experimental artifacts across samples. The assumption is that any sample-specific bias (e.g., in capture or amplification efficiency) affects all genes equally via scaling of the expected mean count for that gene. The size factor for each sample then represents the estimate of the relative bias in that sample, so division of its counts by its size factor should remove that bias.

## Correlation of gene expression across cell types

For a given gene $g$, after performing a $t$ test of its expression between RA and the control group in FLS, we obtain its $t$ statistics denoted $t_{\text{diff-expr}}(g, C)$. Then, we compute the correlation of this score across a cell type pair $(C1, C2)$ as follows:

$$\text{Correction}(C1, C2) = \text{Pearson}\big(\{t_{\text{diff-expr}}(g, C1)_{g \in G}\}, \{t_{\text{diff-expr}}(g, C2)_{g \in G}\}\big). \quad (1)$$

## GRNs in FLS

We inferred our GRNs with PANDA [96] by combining gene expression profiles of FLS [20] with prior knowledge about TF binding motifs (binary) and TF-TF interactions [97, 98]. These were inferred from the StringDB [32] and CIS-BP database [31], and can be downloaded directly from the GRAND database [99] (https://grand.networkmedicine.org/). Briefly, PANDA uses message passing to integrate a prior network (obtained by mapping TF motifs to the genome) with protein–protein interaction and gene expression data by optimizing the weights of edges in the networks with iterative steps. Applied to our data, PANDA produced directed networks of TFs to their TGs, comprising 644 TFs and 18,992 genes, resulting in 12,230,848 edges. Here, each edge between a TF and its TG is associated with a weight, which represents the likelihood of a regulatory interaction between the TF and its TG. The weight values are normalized with a Z-score and range roughly from –3 and 3, which corresponds to how many SD it is below (negative Z-score) or above (positive Z-score) the mean of all other weights in the network.

Then, we used LIONESS [33] to estimate an individual GRN for each sample in the population (18 RA and 12 OA networks). LIONESS estimates sample-specific networks by sequentially leaving each sample out, calculating a network (with PANDA) with and without that sample, and using linear interpolation to estimate the network for the left-out sample. All networks were inferred with python library netZooPy (https://github.com/netZoo/netZooPy). Note that we chose LIONESS because it is specifically designed to estimate sample-specific GRNs, making it well suited for capturing the heterogeneity inherent in diseases like RA. In comparison, other tools such as CARNIVAL [100], DOROTHEA [101], ISMARA [102], LAMP

[103], and CoRegNet [104] provide regulatory network predictions but operate primarily at the level of motifs, which may not offer the same granularity for sample-specific analysis.

## Analysis of TF RA regulatory activity in GRNs

For each individual sample in our RNA-seq data, we established an FLS-directed network of TF nodes with regulatory edges linking them to their TGs, with weight representing the likelihood of the regulatory interaction between the two nodes. We leveraged this collection of networks to test whether the weights of these regulatory edges differed significantly between RA and control tissues, and to identify the TFs driving these regulatory differences. The $t$ test was used for the computation of (i) the differential gene expression between the RA and control group and (ii) the differential weight of the regulatory edges between the RA and control group. The obtained scores are denoted as $t_{\text{diff-expr}}$ and $t_{\text{diff-edge}}$, respectively. We define RA DEGs as the genes having a $t_{\text{diff-expr}} > 1$ in their RA differential expression. Note that from the definition of the $t$ score, these represent genes where the difference between the two phenotype groups is higher than the SD of their gene expression across all samples. Then, we quantified the TF regulatory importance as the average absolute differential weight of the regulatory edges $t_{\text{diff-edge}}$ of the RA targeted genes, where only the TGs listed in the TF motifs were considered (Equation (2)). We expected that TFs with the highest scores would be the most likely to contribute to RA regulation. Defining as the set of all genes in the network, we formalize the computation of the TF scores with

$$\{\text{RA DEGs}\} = \{|t_{\text{diff-expr}}(gene)| > 1, gene \in G\},$$

$$\{\text{TF targets}\} = \{\text{motif}(TF, gene) = \text{true}, gene \in G\},$$

$$T = \{\text{RA DEGs}\} \cap \{\text{TF targets}\},$$

$$\text{TF}_{\text{score}} = \frac{1}{|T|} \sum_{gene \in T} |t_{\text{diff-edge}}(TF, gene)|. \quad (2)$$

## TF-TF co-regulation network

We quantify the co-regulation between TFs by evaluating the Pearson correlation between their common gene target's differential edge weights. TFs with less than 10 common targets are associated with co-regulation of 0. Defining G as the set of all genes in the network, we write

$$T_{ij} = \{\text{motif}(TF_i, gene) = \text{motif}(TF_j, gene) = \text{true}, gene \in G\},$$

$$\text{Co-regulation}(TF_i, TF_j) = \text{Correlation}\big(\{t_{\text{diff-edge}}(TF_i, gene), t_{\text{diff-edge}}(TF_j, gene), gene \in T_{ij}\}\big). \quad (3)$$

## KDA

Two independent lists of RA-associated genes, denoted as the *DEG list* and *Literature list*, were compiled with a DEG meta-analysis [44,

45, 46, 47, 48) and by aggregating together several databases (16, 37, 38, 39, 40, 41, 42, 43), respectively (Table S14), as described in reference 49. Then, 14 networks from different human organs, tissues, and cell types were downloaded (Table S7). Each of these networks and a list of RA-associated genes were used to run a KDA with the Mergeomics R library (36) (for additional details, refer to reference 49).

### Isolation and culture of FLS

Primary FLS cell lines developed from RA patients were obtained as previously described (86, 105). Briefly, synovial tissues were obtained under IRB-approved protocols and all patients signed informed consent forms. Tissues were freshly obtained, minced, and incubated with a solution containing DNase (0.15 mg/ml), hyaluronidase type I-S (0.15 mg/ml), and collagenase type IA (1 mg/ml) (Sigma-Aldrich) in DMEM (Invitrogen) for 1 h at 37°C. Cells were washed and resuspended in complete media containing DMEM supplemented with 10% FBS (Invitrogen), glutamine (300 ng/ml), amphotericin B (250 $\mu$g/ml) (Sigma-Aldrich), and gentamicin (20 $\mu$g/ml) (Invitrogen). After overnight culture, non-adherent cells were removed and adherent cells cultured. All experiments were performed with FLS after passage 4 (>95% FLS purity).

### siRNA knockdown

RA FLS (four to five cell lines from different patients) were transfected with siRNA BACH1 or a non-coding control using the Dharmacon SMARTpool siRNA according to the manufacturer's instructions (Dharmacon, GE Lifesciences) as previously described (86). Cells were then incubated at 37°C for 24–48 h before initiating the functional assays as described below, or using the cells for RNA sequencing. Knockdown was confirmed with qRT-PCR.

### FLS assays

#### Invasion assay
The in vitro invasiveness of FLS was assayed in a Transwell system using Matrigel-coated inserts (BD Biosciences), as previously described (14, 81, 105). Briefly, siRNA-transfected FLS were harvested by trypsin–EDTA digestion and resuspended in 500 $\mu$l of serum-free DMEM. $2 \times 10^4$ cells were placed in the upper compartment of each Matrigel-coated insert. The lower compartment was filled with media containing 10% FBS, and the plates were incubated at 37°C. After 24 h, the upper surface of the insert was wiped with a cotton swab to remove non-invading cells and the Matrigel layer. The opposite side of the insert was stained with crystal violet (Sigma-Aldrich), and the number of cells that invaded through Matrigel was analyzed with ImageJ software. Experiments were done in duplicate.

#### Adhesion to the Matrigel assay
The FLS adhesion assay was done as previously reported (85). Briefly, transfected cells were trypsinized and counted. 6,000 cells per well were plated in triplicate in a 96-well plate previously coated with 5 $\mu$g/ml of Matrigel (BD), in complete media. After 2 h, non-adherent cells were washed out with PBS 1X, and adherent

cells were stained with crystal violet. Cells were manually counted and read with a spectrophotometer at 590 nm.

#### Migration in the wound healing assay
Transfected FLS were briefly trypsinized and counted. 6,000 cells per well were plated in triplicates in a 96-well plate. Cells were allowed to grow until confluence (usually 24 h). Then, a wound (scratch) was created using a 10-$\mu$l pipette tip. Pictures were taken at this initial time (time 0) and 24 h later. FLS migration was determined using ImageJ software by subtracting the density (the number of cells that cross into the wound/scratch area) after 24 h from the density of time 0 (reference point).

#### Proliferation
Transfected FLS were trypsinized and counted. 3,000 cells per well were plated in triplicates in a 96-well plate in complete media with 10% FBS. After the indicated times, cells were incubated with Promega CellTiter 96 AQueous One Solution Cell Proliferation Assay (MTS) (Madison, WI) according to the manufacturer's instructions. Proliferation was assessed by colorimetric reading at 490 nm.

### Immunofluorescence microscopy

Immunofluorescence was performed as previously reported (85). Briefly, siRNA-transfected cells were plated on a glass coverslip with media containing 10% FBS. Cells were fixed with 4% formaldehyde for 15 min at RT and permeabilized with PBS/Triton X-100 0.1% for 5 min. Non-specific binding was blocked with 5% nonfat milk. Cells were then stained with Alexa Fluor 488 (green) phalloidin (Invitrogen) to stain the actin filament and anti-phospho-FAK (pFAK; Abcam), followed by a secondary Alexa Fluor 594 (red) antibody to identify pFAK and lamellipodia. Images were acquired with a Leica DMi8 microscope at 600× magnification and analyzed with Leica Application Suite X (LAS X) software (Leica).

FLS actin filaments were scored following the system described in reference 77. Briefly, actin filaments were categorized into three groups: (i) no filaments visible in the central area of the cell, (ii) some fine filaments present in the central area of the cell, and (iii) more than 90% of cell area filled with thick filaments.

### Matrix metalloproteinase (MMP) quantification in zymograms

Gelatin (MMP2) and casein (MMP3) zymography was performed according to previously described methods (14). Briefly, RA FLS transfected with either siRNA BACH1 or control were cultured on Matrigel and supernatants concentrated with Micron centrifugal filters (MilliporeSigma) and the protein content quantified. The same amount of protein per sample was used in each experiment. Protein was mixed with Tris–glycine–sodium dodecyl sulfate (SDS) sample buffer (Invitrogen), loaded into a zymogram precast gel (Invitrogen), and run for 90 min at 125 V. After electrophoresis, gels were treated with renaturing buffer (Invitrogen), followed by incubation in developing buffer (Invitrogen) at 37°C overnight. Gels were stained with SimplyBlue SafeStain (Invitrogen) for 1 h at RT and washed. Areas of protease activity appeared as clear bands against a dark-blue background.

## RNA extraction and RNA sequencing

Total RNA was extracted and isolated from synovial tissues, quantified by NanoDrop, and 400 ng per sample was sent to Novogene (Beijing, China) for sequencing and analyses.

## Data Availability

All the gene lists obtained in this study, along with the data and the code to reproduce all figures presented in this article, are made available publicly on GitHub at https://github.com/AI-SysBio/RA-drug-discovery.

## Supplementary Information

## Acknowledgements

This research was supported by the COSMIC European Training Network, funded by the European Union's Horizon 2020 research and innovation program under grant agreement No. 765158. PS Gulko and T Laragione were funded by the NIH R01AR07316, by the Icahn School of Medicine at Mount Sinai, and by the Eunice Bernhard fund.

### Author Contributions

A Pelissier: conceptualization, data curation, formal analysis, validation, investigation, visualization, methodology, and writing—original draft, review, and editing.
T Laragione: conceptualization, data curation, investigation, methodology, and writing—original draft, review, and editing.
C Harris: resources, validation, and methodology.
M Rodríguez Martínez: conceptualization, supervision, methodology, and writing—original draft.
PS Gulko: conceptualization, resources, data curation, formal analysis, supervision, funding acquisition, validation, investigation, methodology, project administration, and writing—original draft, review, and editing.

### Conflict of Interest Statement

The authors declare that they have no conflict of interest.

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
