## [Reviewer comments · Life Science Alliance]

Life Science Alliance

BACH1 as a Key Driver in Rheumatoid Arthritis Fibroblast-like Synoviocytes

Aurelien Pelissier, Teresina Laragione, Carolyn Harris, María Rodríguez Martínez, and Percio Gulko

DOI: <https://doi.org/10.26508/lsa.202402808>

Corresponding author(s): Aurelien Pelissier, IBM Research - Zurich

Review Timeline:

Submission Date:	2024-05-06
Editorial Decision:	2024-08-02
Revision Received:	2024-10-08
Editorial Decision:	2024-10-10
Revision Received:	2024-10-16
Accepted:	2024-10-17

Transaction Report:

August 2, 2024

Re: Life Science Alliance manuscript #LSA-2024-02808

Mr. Aurelien Pelissie
IBM Research - Zurich
Bellariastrasse 62
Zurich, Zurich 8038
Switzerland

Dear Dr. Pelissie,

Thank you for submitting your manuscript entitled "Gene Network Analyses Identify Co-regulated Transcription Factors and BACH1 as Key Driver in RA FLS" to Life Science Alliance. The manuscript was assessed by expert reviewers, whose comments are appended to this letter. We invite you to submit a revised manuscript addressing the Reviewer comments.

Thank you for this interesting contribution to Life Science Alliance. We are looking forward to receiving your revised manuscript.

Sincerely,

B. MANUSCRIPT ORGANIZATION AND FORMATTING:

Reviewer #1 (Comments to the Authors (Required)):

Please find my comments below

- When the authors state that BACH1 and HLX is specific to FLS "Among them, 7 were new TF regulators (HLX, BACH1, ETV7, TGIF1, ELF1, HIVEP1, and PLAGL1) not previously implicated in RA, and two of these (BACH1 and HLX) were specific to FLS, while the others were also associated with T and/or B cell signatures. "

While it is not clear to the reader how the authors come to the conclusion that this is specific for FLS and not other cells that are present in RA.

- When the authors explain the network in Figure 3 there are 5 clusters that are identified with their related TFs. In here the cluster with BACH1 is the major cluster. After noting this the authors continue to the following analysis on BACH1 and focus of the TF and genes that are regulated with BACH1. But then the TF are not related to each other of the 2 analysis. It feels like it would be important to link these two TF signatures beside just BACH1.

- When the authors are talking about filament counts in figure 6, there is a category of "some filaments" this is in the figure legend not explained what "some" means. It would be good for the authors to make this more detailed.

Minor:

- Figure 1a would help to clarify that upregulated is associated with RA while down-regulated is with OA samples.

Reviewer #2 (Comments to the Authors (Required)):

Review for the submission: Gene Network Analyses Identify Co-regulated Transcription Factors and BACH1 as a Key Driver in Rheumatoid Arthritis Fibroblast-like Synoviocytes by Aurelien Pelissier et al.

The authors describe their efforts to identify co-regulatory networks of TFs in Rheumatoid Arthritis fibroblasts.

As stated in the introduction, they aim to construct sample-specific gene regulatory networks (GRNs) in both RA and controls.

Q1: What are the advantages of sample-specific GRNs, especially in a disease with so much heterogeneity? What is the applicability of a sample-specific GRN?

The authors use RNAseq data from 18 RA and 12 OA biopsies and compute the differential gene expression of genes between RA and OA samples, followed by an overrepresentation pathway analysis. The paper describing the dataset and the biopsies is not open access -"Defining inflammatory cell states in rheumatoid arthritis joint synovial tissues by integrating single-cell transcriptomics and mass cytometry", by Zhan et al - and it is not easy to understand what data type exactly the authors are using, just by reading the Introduction. They clarify in the Methods section, but it would be helpful for the reader to state it clearly from the beginning.

Q2: Why did the authors choose this dataset among others available in the public space? Is there something particular about it?

Next, the authors used the published LIONESS algorithm to estimate sample-specific regulatory networks.

Q3: Why did the authors choose this method? What are its advantages and disadvantages compared to other tools, such as CARNIVAL/DOROTHEA, ISMARA, LAMP, and CoRegNet?

While the authors state that they infer sample-specific GRNs in contrast to cohort-specific GRNs, it is unclear if, at the end, they produce 18 different GRNs (18 RA samples) or one GRN with edges based on likelihood inferred from estimations in each sample. The phrasing is confusing.

On page 5, the authors write: "To enhance the reliability of the identified TF regulators and dGRN and mitigate variability commonly linked with reconstructed networks [34], we incorporated a set of 14 publicly available networks, sourced from the literature, which also incorporated edge weights as a measure of the interaction confidence between nodes in various cell types

and tissues. Since there are no publicly accessible networks of synovial tissues or FLS, we selected networks derived from immune-related tissues, such as lymph nodes, spleen, tonsils, and blood, as well as various immune cell types (see Supplementary Table S8)."

Q4. The authors seem to ignore the literature on RA networks.

The first comprehensive RA map was published in 2010

<https://pubmed.ncbi.nlm.nih.gov/20419126/>

An updated version was published in 2020

<https://www.ncbi.nlm.nih.gov/pmc/articles/PMC7170216/#ref23>

And an RA cell atlas was published in 2022

<https://www.frontiersin.org/journals/systems-biology/articles/10.3389/fsysb.2022.925791/full>

All these valuable, disease-specific networks of synovial tissues and cells are publicly accessible.

Updated RA FLS networks with calculated scores of specificity can also be found here:

<https://www.nature.com/articles/s41540-023-00294-5>

And here:

<https://journals.plos.org/ploscompbiol/article?id=10.1371/journal.pcbi.1010408>

Q5. The authors state that they incorporated 14 publicly available networks - from where? And with what criteria? The incorporated edge weights? This part needs more clarification and explanation.

Q6. For the Key Driver Analysis (KDA), the authors used a list of RA-associated signatures as a starting point. Where do these signatures come from? Again, why was this method chosen? What is the added value?

Q7. The authors focus solely on BACH1, for the in vitro validation.

It is widely known that changing an approach, tool, or algorithm might drastically change results. How robust is the finding? Did the authors try to see if BACH1 is also present in a different dataset? Confirming some of the findings in independent datasets will add more confidence to the results.

Given the disease heterogeneity and the small dataset, I suggest analyzing at least one more dataset to confirm the findings.

Also, a comparison with previously reported TFs in RA FLS is missing. For example, in <https://www.nature.com/articles/s41467-022-33785-w>

BACH1, FOSL1, and ETV7 are also identified, among others. In Zerrouk et al., the authors identify TFs for RA FLS and for five different subpopulations using transcriptomics, RNAseq, and scRNAseq and three different algorithms. How do the findings compare with those results?

The first part of the manuscript contains steps of network inference, TF activity estimation, and overrepresentation analysis - steps considered standard in network biology. The dataset analyzed is quite limited in sample size, and the choice of the methods is not well justified (in comparison with the state of the art). The authors use datasets, methods, and tools without describing what motivated them to use the particular dataset, algorithm, or methodology and its added value. The authors do not compare or discuss their findings with published literature using similar approaches focused on RA FLS.

In the discussion, the authors state:

Gene regulatory processes vary across cell types, and only a limited number of studies have characterized RA drivers in cell types relevant to RA pathogenesis, such as FLS [11, 68].

Again, in Zerrouk et al., the authors identify TFs for RA FLS and for five different subpopulations using transcriptomics, RNAseq, scRNAseq, and three different algorithms. How do the findings compare with those results?

HIF1A is also identified as a metabolic switch in RA FLS in

<https://journals.plos.org/ploscompbiol/article?id=10.1371/journal.pcbi.1010408>

<https://insight.jci.org/articles/view/179392>

A discussion highlighting what has already been reported and what is novel is missing. Overall, this contribution can be strengthened by adding justification and comparison with the state of the art, both for the methodological choices and the reported findings. A step-by-step explanation of the methodology would significantly improve comprehension, as it is challenging to follow all the steps. A figure describing the pipeline would also be welcome.

Reviewer #3 (Comments to the Authors (Required)):

In this paper by Pelissier et al., a new bioinformatics analysis of published data RA FLS expression profiles has identified relevant transcription factors and their target genes. By analysing sample-specific gene expression data, they constructed gene regulatory networks (GRNs) to distinguish RA from OA patients. Furthermore, ex vivo siRNA-mediated knockdown of the BACH1 transcription factor in RA FLS cell lines affected FLS adhesion, migration, and disrupted normal actin fiber and lamellipodia formation.

Major revisions:

1. The authors state in the abstract, "Six of these TFs are new and have not been previously implicated in RA, including BACH1,

HLX, and TGIF1," and in the discussion, they mention "seven not previously implicated in RA (BACH1, HLX, ETV7, TGIF1, ELF1, HIVEP1, and PLAGL1)." BACH1 has been linked with arthritis before as referenced in Armaka et al. 2022 paper in Genome Medicine. This study identified BACH1 GRN as active in pathogenic synovial fibroblast subpopulations in arthritis and was shown to be conserved in human RA synovial fibroblasts.

2. The analysis is based on data from Fan Zhang et al study, "Defining inflammatory cell states in rheumatoid arthritis joint synovial tissues by integrating single-cell transcriptomics and mass cytometry" (Nature Immunology, 2019), where RA patients are classified into leukocyte-rich and leukocyte-poor groups. Further analysis to determine if these GRNs are more prevalent in leukocyte-rich compared to leukocyte-poor patients would be insightful. It would also be interesting to see if patient stratification could be conducted based on differential GRNs.

3. In the ex vivo siRNA-mediated knockdown experiments of BACH1 in RA FLS cell lines, BACH1 levels should be presented to confirm the downregulation of BACH1 expression.

4. The authors state in the abstract that "The main BACH1 targets included those implicated in fatty acid metabolism and ferroptosis," and in the results section pathway enrichment analysis showed an over-representation of "fatty acid degradation" and "ferroptosis" pathways. However, validation experiments on these pathways need to be performed in bach1-knockdown RA FLS and their respective controls e.g. iron measurements, cell viability assays, lipid metabolism, and peroxidation assays.

5. BACH1 is implicated in the oxidative stress response. Did reducing BACH1 levels influence the oxidative stress status of RA FLS?

6. BACH1 has been shown to promote osteoclastogenesis in osteoarthritis (OA) in vivo, confirmed ex vivo through an osteoclastogenesis assay using osteoclasts derived from bone marrow-derived macrophages. Does knockdown of BACH1 in RA FLS cell lines influence RANKL protein levels?

7. The role of BACH1 in FLS behaviour, such as adhesion, migration, actin fiber, and lamellipodia formation, should be validated by ex vivo knockdown experiments in primary FLS from human RA patients to confirm findings shown in RA patient FLS cell lines.

The aforementioned additional experiments could be conducted within 3 months.

Reviewer 1:

- When the authors state that BACH1 and HLX is specific to FLS "Among them, 7 were new TF regulators (HLX, BACH1, ETV7, TGIF1, ELF1, HIVEP1, and PLAGL1) not previously implicated in RA, and two of these (BACH1 and HLX) were specific to FLS, while the others were also associated with T and/or B cell signatures." While it is not clear to the reader how the authors come to the conclusion that this is specific for FLS and not other cells that are present in RA.

We added a sentence clarifying this:

"By integrating our results with those from our parallel study on other cell types, which employed a similar methodology, we found that only BACH1 and HLX were specific to FLS, while the other factors were also associated with T and/or B cell signatures."

- When the authors explain the network in Figure 3 there are 5 clusters that are identified with related TFs. In here the cluster with BACH1 is the major cluster. After noting this the authors continue to the following analysis on BACH1 and focus of the TF and genes that are regulated with BACH1. But then the TF are not related to each other of the 2 analysis. It feels like it would be important to link these two TF signatures beside just BACH1.

We added a sentence discussing the link between BACH1 targets and its coregulatory clusters

"Interestingly, RUNX1 and CFBF were also identified as significant TF drivers in RA FLS, but in a distinct co-regulatory cluster, with only moderate correlations to BACH1 (0.31 and 0.16, respectively). This suggests that, although these two TFs are targets of BACH1, they exert control over their respective targets independently of BACH1 regulation."

- When the authors are talking about filament counts in figure 6, there is a category of "some filaments" this is in the figure legend not explained what "some" means. It would be good for the authors to make this more detailed.

Actin filaments were categorized into three groups (i) no filaments visible in the central area of the cell, (ii) some fine filaments present in the central area of the cell and (iii) more than 90% of cell area filled with thick filaments. We added this information to the main text.

Minor:

- Figure 1a would help to clarify that upregulated is associated with RA while down-regulated is with OA samples.

We adjusted this information in the caption of Figure 1A.

Reviewer 2:

The authors describe their efforts to identify co-regulatory networks of TFs in Rheumatoid Arthritis fibroblasts. As stated in the introduction, they aim to construct sample-specific gene regulatory networks (GRNs) in both RA and controls.

Q1: What are the advantages of sample-specific GRNs, especially in a disease with so much heterogeneity? What is the applicability of a sample-specific GRN?

The heterogeneity in RA is precisely why we construct sample-specific GRNs. These networks capture unique regulatory dynamics for each sample, preserving the variability. This allows for statistical testing across constructed networks, helping identify patterns while adapting to individual differences. We added a sentence in the introduction to clarify this.

The authors use RNAseq data from 18 RA and 12 OA biopsies and compute the differential gene expression of genes between RA and OA samples, followed by an overrepresentation pathway analysis. The paper describing the dataset, and the biopsies is not open access -"Defining inflammatory cell states in rheumatoid arthritis joint synovial tissues by integrating single-cell transcriptomics and mass cytometry", by Zhan et al - and it is not easy to understand what data type exactly the authors are using, just by reading the Introduction. They clarify in the Methods section, but it would be helpful for the reader to state it clearly from the beginning.

Q2: Why did the authors choose this dataset among others available in the public space? Is there something particular about it?

Zhan et al.'s dataset was selected because, at the time of the study, it was the only available resource providing FLS specific bulk RNA-seq data for both RA and OA in the synovial tissues of human. We have added a sentence in the introduction to clarify this point and included additional details about the nature of the dataset.

"In this study, we provide a comprehensive analysis of gene regulation in RA FLS. We utilized a cell-type-specific bulk RNA-seq dataset of FLS from patients with RA (n = 18) and OA controls (n = 13)[20] to construct sample-specific gene regulatory networks (GRNs) for both RA and control samples, generating 18 RA and 12 OA FLS networks. Employing sample-specific GRNs is essential for capturing the heterogeneity of RA, as it tailors the networks to each individual sample. This contrasts with previous studies, which focused on cohort-specific GRNs[26, 27, 7], allowing us to preserve inter-sample variability. Through differential analysis of the network edges, we identified key transcription factors driving FLS-specific expression differences, offering new insights into RA."

Q3. Next, the authors used the published LIONESS algorithm to estimate sample-specific regulatory networks.

Why did the authors choose this method? What are its advantages and disadvantages compared to other tools, such as CARNIVAL/DOROTHEA, ISMARA, LAMP, and CoRegNet?

We chose the LIONESS algorithm because it is specifically designed to estimate sample-specific gene regulatory networks, making it well-suited for capturing the heterogeneity inherent in diseases

like RA. In comparison, other tools like CARNIVAL and DOROTHEA focus more on downstream transcription factor activity predictions rather than building sample-specific networks. ISMARA provides regulatory network predictions but operates primarily at the level of motifs, which may not offer the same granularity for sample-specific analysis. Finally, LAMP and CoRegNet focus on network inference but are less tailored to sample-specific estimations, often generating networks for groups of samples rather than individuals.

We have justified our choice in the introduction and method section.

While the authors state that they infer sample-specific GRNs in contrast to cohort-specific GRNs, it is unclear if, at the end, they produce 18 different GRNs (18 RA samples) or one GRN with edges based on likelihood inferred from estimations in each sample. The phrasing is confusing.

We inferred 18RA and 12 OA GRNs. We added details in the result and method section.

Q4. On page 5, the authors write: "To enhance the reliability of the identified TF regulators and dGRN and mitigate variability commonly linked with reconstructed networks [34], we incorporated a set of 14 publicly available networks, sourced from the literature, which also incorporated edge weights as a measure of the interaction confidence between nodes in various cell types and tissues. Since there are no publicly accessible networks of synovial tissues or FLS, we selected networks derived from immune-related tissues, such as lymph nodes, spleen, tonsils, and blood, as well as various immune cell types (see Supplementary Table S8)."

The authors seem to ignore the literature on RA networks. The first comprehensive RA map was published in 2010 <https://pubmed.ncbi.nlm.nih.gov/20419126/>. An updated version was published in 2020 <https://www.ncbi.nlm.nih.gov/pmc/articles/PMC7170216/#ref23> and an RA cell atlas was published in 2022 <https://www.frontiersin.org/journals/systems-biology/articles/10.3389/fsysb.2022.925791/full>. All these valuable, disease-specific networks of synovial tissues and cells are publicly accessible. Updated RA FLS networks with calculated scores of specificity can also be found here: <https://www.nature.com/articles/s41540-023-00294-5>, and here: <https://journals.plos.org/ploscompbiol/article?id=10.1371/journal.pcbi.1010408>

In the KDA study, we did not specifically focus on an RA network. Instead, we started with a global, general network and then incorporated RA-specific markers through the key driver analysis. This approach was chosen because our goal was to discover new genes or potential pathways, rather than relying solely on existing RA prior knowledge. Furthermore, the networks mentioned above are qualitative (or Boolean) and represent curated maps of known pathways, whereas our KDA approach requires quantitative inferred likelihood GRNs.

Still, we acknowledge that this section lacked sufficient detail, so we made changes to this section to add more details.

Q5. The authors state that they incorporated 14 publicly available networks - from where? And with what criteria? The incorporated edge weights? This part needs more clarification and explanation.

Since RA is an autoimmune disease, we incorporated cell-type specific networks from immune cells that are involved in RA. The networks were sourced from the GIANT project, which provides networks linking genes to each other based on the likelihood of their interaction. This approach allowed us to capture potential gene interactions relevant to each immune cell implicated in RA. We added explanation in the text.

“Relying on the predictions of a single computational method might lack the robustness required to identify promising therapeutic targets. To increase our confidence in the identified RA regulators, we augmented our study by incorporating a selection of pre-existing literature-derived networks, which also included edge weights as a metric for assessing the confidence of the interactions between nodes. These include RIMBANET [34], StringDB [32], and GIANT [35], a collection of networks that accurately capture tissue-specific and cell type-specific functional interactions. As RA is an autoimmune disease, we selected networks computed from immune-related tissues and cell types (including lymph nodes, spleen, tonsils, and blood). In summary, 14 additional networks were collected for our analysis, as detailed in the Supplementary Table S8.”

Q6. For the Key Driver Analysis (KDA), the authors used a list of RA-associated signatures as a starting point. Where do these signatures come from? Again, why was this method chosen? What is the added value?

We clarified in the text and changed the paragraph (See paragraph above.)

“While these networks recapitulate general immune knowledge derived from various data types, they are not specific to synovial tissues. Therefore, they are unable to discern RA-specific relationships between TFs and TGs as effectively as the PANDA framework does. We hence designed a different approach based on the key driver analysis (KDA) [36], a computational pipeline to uncover major disease-associated regulators or causative hubs in a biological network (Methods Section 2.6). Briefly, genes exhibiting more connections to RA-associated genes than expected by random chance were considered potential drivers. KDA analysis requires the definition of RA-associated signatures, i.e. lists of genes associated with the disease. Here, we compiled known RA-associated genes from the literature, including GWAS [37, 16, 38], knowledge-based datasets [39, 40, 41], and known drug targets [42, 43], as well as prior meta-studies and datasets [44, 45, 46, 47, 48] (Methods Section 2.6). Using this list of RA-associated signatures as a starting point, we obtained a list of 174 key TF drivers that were identified as a key driver gene (KDGs) in at least one of the 14 networks (Supplementary Table S9).”

Q7. The authors focus solely on BACH1, for the in vitro validation. It is widely known that changing an approach, tool, or algorithm might drastically change results.

How robust is the finding? Did the authors try to see if BACH1 is also present in a different dataset? Confirming some of the findings in independent datasets will add more confidence to the results. Given the disease heterogeneity and the small dataset, I suggest analyzing at least one more dataset to confirm the findings.

In addition to being a significant TF in our FLS differential network, BACH1 was also identified as a key driver with the KDA approach, as outlined in section 1.3, "An independent approach increases the confidence in the identified regulators." The main objective of this section was to enhance the robustness of the identified TFs. Most of the TFs we identified through these two combined approaches have already been reported in the literature, further validating the reliability of our method.

Also, a comparison with previously reported TFs in RA FLS is missing. For example, in <https://www.nature.com/articles/s41467-022-33785-w>, BACH1, FOSL1, and ETV7 are also identified, among others. In Zerrouk et al., the authors identify TFs for RA FLS and for five different subpopulations using transcriptomics, RNAseq, and scRNAseq and three different algorithms. How do the findings compare with those results?

The first part of the manuscript contains steps of network inference, TF activity estimation, and overrepresentation analysis - steps considered standard in network biology. The dataset analyzed is quite limited in sample size, and the choice of the methods is not well justified (in comparison with the state of the art). The authors use datasets, methods, and tools without describing what motivated them to use the particular dataset, algorithm, or methodology and its added value.

The authors do not compare or discuss their findings with published literature using similar approaches focused on RA FLS. In the discussion, the authors state: Gene regulatory processes vary across cell types, and only a limited number of studies have characterized RA drivers in cell types relevant to RA pathogenesis, such as FLS [11, 68]. Again, in Zerrouk et al., the authors identify TFs for RA FLS and for five different subpopulations using transcriptomics, RNAseq, scRNAseq, and three different algorithms. How do the findings compare with those results? HIF1A is also identified as a metabolic switch in RA FLS in

<https://journals.plos.org/ploscompbiol/article?id=10.1371/journal.pcbi.1010408> and <https://insight.jci.org/articles/view/179392>

A discussion highlighting what has already been reported and what is novel is missing. Overall, this contribution can be strengthened by adding justification and comparison with the state of the art, both for the methodological choices and the reported findings. A step-by-step explanation of the methodology would significantly improve comprehension, as it is challenging to follow all the steps. A figure describing the pipeline would also be welcome.

Thank you for pointing this out. We would like to note that the studies mentioned did not investigate networks, drivers, or the specific effects of BACH1, but rather listed BACH1 among differentially expressed genes (DEGs) or as epigenetically regulated. Still, while we briefly discussed our finding in

the context of other similar studies, we agree that comparison to previous work was incomplete, and we added two full paragraphs to the discussions.

“Using sample-specific GRNs was crucial for capturing the heterogeneity of RA, as it allowed the networks to be tailored to each individual sample, thus contrasting to previous studies on FLS, that predominantly focused on cohort-specific GRNs [11, 18, 82, 83]. Additionally, we directly analyzed FLS from human synovial tissue biopsies, whereas these prior studies did not, potentially offering new insights into RA mechanisms.”

“Importantly, although our methodology differs significantly from previous studies on RA FLS [11, 18, 82, 83], both in terms of methods and datasets, many of the significant TF we identified were also highlighted in those works. Ainsworth et al. [11], who utilized epigenetic data and generated their own FLS cell line datasets, identified BACH1, FOSL1, and ETV7, consistent with our results. Armaka et al. [83] derived their network from sc-RNAseq data on mice and mapped it to human data, with RUNX1 and BACH1 emerging as shared drivers. Zerrouk et al. [18] focused on differentially expressed genes (DEGs) from RA versus OA synovial tissues, though this approach is limited by the known cellular composition differences between these conditions [49]. Nevertheless, they identified STAT1 and IRF4, two prominent RA transcription factors, which we also found. Lastly, Aghakhani et al. [82] constructed a qualitative Boolean network based on prior RA FLS knowledge and, like us, identified HIF1. By integrating our findings with these studies, we can confidently assert that transcription factors such as BACH1, FOSL1, ETV7, RUNX1, STAT1, IRF4, and HIF1 represent promising therapeutic targets. Notably, our work provides a more in-depth analysis of BACH1, whereas the other studies only briefly mention it as a potential driver.”

Reviewer 3:

In this paper by Pelissier et al., a new bioinformatics analysis of published data RA FLS expression profiles has identified relevant transcription factors and their target genes. By analyzing sample-specific gene expression data, they constructed gene regulatory networks (GRNs) to distinguish RA from OA patients. Furthermore, ex vivo siRNA-mediated knockdown of the BACH1 transcription factor in RA FLS cell lines affected FLS adhesion, migration, and disrupted normal actin fiber and lamellipodia formation.

Major revisions:

1. The authors state in the abstract, "Six of these TFs are new and have not been previously implicated in RA, including BACH1, HLX, and TGIF1," and in the discussion, they mention "seven not previously implicated in RA (BACH1, HLX, ETV7, TGIF1, ELF1, HIVEP1, and PLAGL1)." BACH1 has been linked with arthritis before as referenced in Armaka et al. 2022 paper in Genome Medicine. This study

identified BACH1 GRN as active in pathogenic synovial fibroblast subpopulations in arthritis and was shown to be conserved in human RA synovial fibroblasts.

We corrected into “have not been previously implicated in RA through ex vivo or in vivo studies.”

2. The analysis is based on data from Fan Zhang et al study, "Defining inflammatory cell states in rheumatoid arthritis joint synovial tissues by integrating single-cell transcriptomics and mass cytometry" (Nature Immunology, 2019), where RA patients are classified into leukocyte-rich and leukocyte-poor groups. Further analysis to determine if these GRNs are more prevalent in leukocyte-rich compared to leukocyte-poor patients would be insightful. It would also be interesting to see if patient stratification could be conducted based on differential GRNs.

While we recognize the importance of FLS heterogeneity in RA, this paper focuses on identifying transcription factors (TFs) that are common across all RA FLS, rather than delving into the characterization of FLS heterogeneity. We have added a paragraph at the end of the discussion to address this point.

“Finally, An intriguing aspect that we did not address in this study is the heterogeneous pathophysiological background of RA [11, 7]. There is considerable evidence of synovial tissue heterogeneity in RA, potentially contributing to the difference in therapeutic response to bDMARDs therapy across patients [93]. Future studies may be able to leverage our method to infer sample-specific GRNs to distinguish between the GRNs of different FLS subtypes. As FLS have emerged as a leading contender for the source of this RA heterogeneity [94, 10], such an approach might provide crucial insights into the regulation behind the observed heterogeneity of therapeutic responses in RA patients.”

3. In the ex vivo siRNA-mediated knockdown experiments of BACH1 in RA FLS cell lines, BACH1 levels should be presented to confirm the downregulation of BACH1 expression.

BACH1 levels were provided in Supplementary Table S13, but we also have included this information in the text.

4&5. The authors state in the abstract that "The main BACH1 targets included those implicated in fatty acid metabolism and ferroptosis," and in the results section pathway enrichment analysis showed an over-representation of "fatty acid degradation" and "ferroptosis" pathways. However, validation experiments on these pathways need to be performed in bach1-knockdown RA FLS and their respective controls e.g. iron measurements, cell viability assays, lipid metabolism, and peroxidation assays. Also, BACH1 is implicated in the oxidative stress response. Did reducing BACH1 levels influence the oxidative stress status of RA FLS?

We appreciate the reviewer’s suggestions. Although our study identified key BACH1 targets related to fatty acid metabolism and ferroptosis, we have not performed those additional validation experiments such as iron measurements, cell viability assays, oxidative stress assessments, or lipid

metabolism and peroxidation assays in BACH1-knockdown RA FLS. We acknowledge the importance of these validations and plan to include them in future studies to better understand BACH1's role in these processes. However, it is worth noting that BACH1 has already been implicated in both ferroptosis and fatty acid metabolism (<https://pubmed.ncbi.nlm.nih.gov/36670112/>) as well as oxidative stress (<https://pubmed.ncbi.nlm.nih.gov/34605540/>).

6. BACH1 has been shown to promote osteoclastogenesis in osteoarthritis (OA) in vivo, confirmed ex vivo through an osteoclastogenesis assay using osteoclasts derived from bone marrow-derived macrophages. Does knockdown of BACH1 in RA FLS cell lines influence RANKL protein levels?

TNFSF11 (which encodes the RANKL protein) did not show any significant changes following BACH1 knockdown in FLS. However, we note that NFATC1, a key transcription factor activated downstream of RANKL-RANK signaling and involved in osteoclastogenesis, was identified as a significant driver TF in FLS. Since this manuscript focuses on FLS and not myeloid-derived cells like osteoclasts, we believe it is unnecessary to further elaborate on osteoclast-related signaling.

7. The role of BACH1 in FLS behaviour, such as adhesion, migration, actin fiber, and lamellipodia formation, should be validated by ex vivo knockdown experiments in primary FLS from human RA patients to confirm findings shown in RA patient FLS cell lines. The aforementioned additional experiments could be conducted within 3 months.

The role of BACH1 in FLS behavior, including adhesion, migration, actin fiber organization, and lamellipodia formation, was in fact studied in primary FLS cell lines established from RA patients, thus validating our discoveries through our knockdown experiments, as detailed in Section 1.7: *Effect of BACH1 Knockdown on FLS Migration, Adhesion, Lamellipodia Formation, and Cell Morphology*. Therefore, the experiments have already been completed and were in the manuscript. We have now added a sentence clarifying that all experiments were done with primary FLS cell lines developed from RA patients.

October 10, 2024

RE: Life Science Alliance Manuscript #LSA-2024-02808R

Mr. Aurelien Pelissie
IBM Research - Zurich
Bellariastrasse 62
Zurich, Zurich 8038
Switzerland

Dear Dr. Pelissie,

Thank you for submitting your revised manuscript entitled "BACH1 as a Master Regulator in Rheumatoid Arthritis Fibroblast-like Synoviocytes". We would be happy to publish your paper in Life Science Alliance pending final revisions necessary to meet our formatting guidelines.

- please be sure that the authorship listing and order is correct
- please upload your manuscript text as an editable doc file
- please upload both your main and supplementary figures as single files
- please upload any table files as editable doc or excel files
- please consult our manuscript preparation guidelines <https://www.life-science-alliance.org/manuscript-prep> and make sure your manuscript sections are in the correct order
- please use the [10 author names, et al.] format in your references (i.e. limit the author names to the first 10)

Figure Check:

- please add scale bars to Figure 6 B&C

LSA now encourages authors to provide a 30-60 second video where the study is briefly explained. We will use these videos on social media to promote the published paper and the presenting author (for examples, see <https://docs.google.com/document/d/1-UWCfbE4pGcDdcgzcmiuJI2XMBJnxKYeqRvLLrLS08s/edit?usp=sharing>). Corresponding or first-authors are welcome to submit the video. Please submit only one video per manuscript. The video can be emailed to contact@life-science-alliance.org

A. FINAL FILES:

B. MANUSCRIPT ORGANIZATION AND FORMATTING:

Sincerely,

October 17, 2024

RE: Life Science Alliance Manuscript #LSA-2024-02808RR

Mr. Aurelien Pelissier
IBM Research - Zurich
Bellariastrasse 62
Zurich, Zurich 8038
Switzerland

Dear Dr. Pelissier,

Thank you for submitting your Research Article entitled "BACH1 as a Key Driver in Rheumatoid Arthritis Fibroblast-like Synoviocytes". It is a pleasure to let you know that your manuscript is now accepted for publication in Life Science Alliance. Congratulations on this interesting work.

DISTRIBUTION OF MATERIALS:

Again, congratulations on a very nice paper. I hope you found the review process to be constructive and are pleased with how the manuscript was handled editorially. We look forward to future exciting submissions from your lab.

Sincerely,
